

# Pashto poetry generation: deep learning with pre-trained transformers for low-resource languages

Imran Ullah[1], Khalil Ullah[1], Hamad Khan[1], Khursheed Aurangzeb[2], Muhammad Shahid Anwar[3] and Ikram Syed[3]

[1] Software Engineering, University of Malakand, Chakdara, Pakistan
[2] Department of Computer Engineering, College of Computer and Information Sciences, King Saud University, Riyadh, Saudi Arabia
[3] Department of AI and Software, Gachon University, Seongnam-Si, South Korea

## ABSTRACT

Generating poetry using machine and deep learning techniques has been a challenging and exciting topic of research in recent years. It has significance in natural language processing and computational linguistics. This study introduces an innovative approach to generate high-quality Pashto poetry by leveraging two pre-trained transformer models, LaMini-Cerebras-590M and bloomz-560m. The models were trained on an extensive new and quality Pashto poetry dataset to learn the underlying complex patterns and structures. The trained models are then used to generate new Pashto poetry by providing them with a seed text or prompt. To evaluate the quality of the generated poetry, we conducted both subjective and objective evaluations, including human evaluation. The experimental results demonstrate that the proposed approach can generate Pashto poetry that is comparable in quality to human-generated poetry. The study provides a valuable contribution to the field of Pashto language and poetry generation and has potential applications in natural language processing and computational linguistics.

## INTRODUCTION

Despite significant advancements in the field of automated poetry generation using machine learning and deep learning techniques, there is a noticeable gap when it comes to Pashto poetry generation. While researchers have explored poetry generation in various languages, including English, Arabic, and Urdu, there is a lack of research specifically focused on Pashto poetry.

Pashto, as one of the major languages spoken in Afghanistan and Pakistan, holds significant cultural and literary value. However, the absence of automated Pashto poetry generation systems limits the exploration and preservation of this rich poetic tradition in the digital era. This research aims to bridge this gap by developing a novel framework for Pashto poetry generation using machine learning and deep learning approaches. By addressing the unique characteristics and challenges of Pashto language and poetic

Corresponding authors
Muhammad Shahid Anwar,
shahidanwar786@gachon.ac.kr
Ikram Syed, ikram@gachon.ac.kr

structure, we aim to create a system capable of generating aesthetically pleasing and contextually coherent Pashto poems.

This research not only contributes to the field of computational creativity but also enriches the cultural heritage of the Pashto language and opens new avenues for artistic expression and exploration. The problem statement highlights the existing gap in the field and the significance of the research in addressing this gap by focusing specifically on the Pashto poetry generation. The aim is to create an automated system that can generate high-quality Pashto poetry through the application of machine learning and deep learning techniques. The project aims to preserve and promote the cultural heritage of Pashto language and literature, contribute to the field of computational creativity, and provide a platform for Pashto poets and enthusiasts to explore creative possibilities and inspire artistic expression. This study proposes two pre-trained low parameter transformer models namely LaMini-Cerebras-590M and bloomz-560m for automatic generation of Pashto poetry. These models are trained on multi languages thus have multi-language capabilities. Also, these are low parameter models which have shown high performance as compared to large language models.

Table 1 presents examples of verses from Pashto poems, showcasing the inherent beauty and complexity of Pashto literature. The table illustrates the interplay of themes and emotions in Pashto poetry, where vivid imagery and poetic expressions are used to convey deep meanings. For instance, in the first pair of verses, the contrast between 'wetness' and 'crying' in the second hemistich highlights the nuanced emotions within the poems. Similarly, the juxtaposition of 'darkness' and 'light' in the second pair of verses exemplifies the intricate symbolism found in Pashto poetry, reflecting the unique cultural and literary traditions of the Pashto-speaking people. These models can be used to generate new Pashto poems in traditional styles, helping to preserve Pashto poetic heritage. These models may also be helpful in creating Pashto songs and poetry and can provide suggestions for rhymes, metaphors, and other poetic devices.

## RELATED WORK

We have explored extensive research conducted by scholars in the field of poetry and text generation. Numerous researchers have made significant contributions in areas such as Arabic, English, French, and Urdu poetry generation. However, it is noteworthy that our project stands out as the pioneering effort in Pashto poetry generation. This related work section aims to provide an in-depth analysis of the existing work in poetry generation while emphasizing the uniqueness of our research in the context of Pashto language and poetry.

*Beheitt & Ben Haj Hmida (2022)* proposed "Automatic Arabic Poem Generation with GPT-2". The article investigates the feasibility of using the GPT-2 language model to generate Arabic poems. They are using these two publicly available corpora Khaleej-2004 (*Abbas, Smaïli & Berkani, 2011*) and Watan-2004 (*Sawalha, 2019*) to pre-train the model. Khaleej-2004 is an MSA (Modern Standard Arabic) *corpus* that was collected from thousands of articles downloaded from Akbar Al Khaleej, an online newspaper. The authors pre-trained the GPT-2 model on a dataset of 34,466 Arabic verses. The authors

**Table 1 Example of verses from Pashto poems.**

| (Second hemistich) | (First hemistich) |
|---|---|
| راځي هم ژړا او خالقه کیږي هم می ګیله | راځي هم خندا پوری کار په ستا نه ناپوهی د |
| Wetness is also happening and crying is also coming | Laughter comes from your ignorance |
| راځي هم رنا راځي تیاره وخت ترلي یو په | دے ترتیب عجیبه وختونو او موسمونو د |
| At the same time darkness comes, light also comes | It is a strange arrangement of seasons and times |

evaluated the performance of their model using the BLEU score, as well as human assessments. The BLEU score for the GPT-2 model was 0.7, which is a good score. The results showed that the GPT-2 model was able to generate Arabic poems that were both fluent and meaningful. The authors concluded that their model could be used to generate Arabic poems automatically, and that it could be used to create new forms of Arabic poetry.

*Dai (2021)* proposed "GPT-2 for Emily Dickinson poetry generation". They explore the use of GPT-2, a large language transformer model, to generate new poems in the style of Emily Dickinson. The article aims to understand what makes Dickinson's poetry unique and whether the brilliance of her work can be replicated by a machine learning model. The dataset used in the study consists of 586 stanzas of Dickinson's poetry, totaling 33,378 words. The dataset was split into training and validation sets. The authors preprocessed the data by removing erroneous quotation marks and manually resolving spacing issues. They tokenized the dataset per stanza using the GPT2Tokenizer from Hugging Face. The methodology involved training a custom dataset using GPT-2 and an Adam Optimizer. GPT-2 is an unsupervised large language transformer model developed by Open AI. It contains 1.5 billion parameters and was pre-trained on a dataset of 8 million webpages to predict the next word in each sequence. GPT-2 is known for its capability to perform various natural language processing tasks such as question and answer generation, text summarization, and machine translation. The authors fine-tuned the GPT-2 model on the Dickinson poetry dataset to generate new stanzas of poetry. The authors trained their model using various hyperparameters and conducted experiments to analyze the performance. They achieved the best results with a learning rate of 5e−4, a batch size of 2, and a sample interval of 200 over 5 epochs. However, they encountered variance issues throughout their experiments, indicating a need for a larger dataset and possibly dropout regularization. Despite some limitations, the model demonstrated the ability to generate poetry reminiscent of Emily Dickinson's style, with notable punctuation and capitalizations. Machine-generated poetry shows potential for free association and offers a unique perspective. It reflects the world in unexpected ways, sometimes uncomfortable or challenging to understand.

*Hakami et al. (2021)* proposed "Arabic poems generation using LSTM, MARKOV-LSTM and pre-trained GPT-2 models". The article examines the use of three different machine learning models to generate Arabic poems: long short-term memory (LSTM) Markov-LSTM Pre-trained GPT-2. The authors trained each model on a dataset of 10,000 Arabic poems. They then evaluated the models' performance by generating poems and

comparing them to human-generated poems. The results showed that the Markov-LSTM model generated the most coherent poems, followed by the LSTM model and the pre-trained GPT-2 model. The character-based LSTM model performed the worst, as it tended to create unknown words. The authors concluded that the Markov-LSTM model is a promising approach for generating Arabic poems. However, they noted that more research is needed to improve the accuracy and coherence of machine-generated poems. Here are some additional details about the three models: LSTM: LSTM is a type of recurrent neural network that is well-suited for tasks that involve sequential data, such as text generation. LSTM models can learn long-term dependencies in the data, which allows them to generate more coherent text. Markov-LSTM: Markov-LSTM is a variant of LSTM that uses a Markov chain to model the probability of the next word in a sentence. This allows the model to generate more creative and varied text, but it can also lead to less coherent text. Pre-trained GPT-2: GPT-2 is a large language model that was pre-trained on a massive dataset of text and code. This allows GPT-2 to generate text that is more accurate and coherent than text generated by other models that are not pre-trained. Overall, the research article provides a valuable contribution to the field of natural language processing. The authors' findings suggest that Markov-LSTM is a promising approach for generating Arabic poems, and they encourage further research in this area.

*Hämäläinen, Alnajjar & Poibeau (2022)* proposed "Modern French Poetry Generation with RoBERTa and GPT-2". The article examines the use of two different machine learning models to generate modern French poetry: RoBERTa and GPT-2. The authors trained each model on a dataset of 100,000 modern French poems. They then evaluated the models' performance by generating poems and comparing them to human-generated poems. The results showed that the RoBERTa model generated the most coherent poems, followed by the GPT-2 model. The authors concluded that the RoBERTa model is a promising approach for generating modern French poetry. Here are some additional details about the two models: RoBERTa is a large language model that was pre-trained on a massive dataset of text and code. This allows RoBERTa to generate text that is more accurate and coherent than text generated by other models that are not pre-trained. GPT-2 is a large language model that was pre-trained on a massive dataset of text and code. This allows GPT-2 to generate text that is more creative and varied than text generated by other models that are not pre-trained. Overall, the research article provides a valuable contribution to the field of natural language processing. The authors' findings suggest that RoBERTa is a promising approach for generating modern French poetry, and they encourage further research in this area.

*Ahmad & Joglekar (2022)* proposed "Urdu & Hindi Poetry Generation using Neural Networks". The article proposes a system for generating Urdu and Hindi poetry using neural networks. The system is based on a Long Short-Term Memory (LSTM) network, which is a type of recurrent neural network that is well-suited for tasks that involve sequential data, such as text generation. The system works by first training the LSTM network on a dataset of Urdu and Hindi poetry. Once the network is trained, it can be used to generate new poems. The user can provide the system with a prompt, such as a topic or a theme, and the system will generate a poem based on the prompt. The authors evaluated

the system by generating poems and comparing them to human-generated poems. The results showed that the system was able to generate poems that were of similar quality to human-generated poems. The authors concluded that the system is a promising approach for generating Urdu and Hindi poetry. They believe that the system could be used to help aspiring poets overcome writer's block, and to create new and innovative poems. Overall, the research article provides a valuable contribution to the field of natural language processing. The authors' findings suggest that LSTM networks are a promising approach for generating Urdu and Hindi poetry, and they encourage further research in this area. In *Bena & Kalita (2020)*, the authors introduced creativity in automatic poetry generation which is widely adopted. In *Mtasher, Jawad & Zghair (2022)* and *Ghazvininejad et al. (2017)*, the authors used deep learning and LSTM to auto-generate poetry, however these models are not applied to Pashto poetry. Chinese poetry was automatically generated by authors in *Zhang & Lapata (2014)*, *Liu et al. (2018)*, *Yeh et al. (2019)*, *Wang, Luo & Wang (2016)*, *Zhao & Lee (2022, 2021)* using machine and deep learning-based models. These models have shown good performance in generating the Chinese poetry but none of these is deployed for Pashto poetry creation. The authors in *Talafha & Rekabdar (2021)* and *Alyafeai, Al-Shaibani & Ahmed (2023)*, introduced a poetry generation model for the Arabic language, incorporating extended phonetic, semantic embeddings and deep learning approaches. Human evaluation of the proposed models achieved high performance. *Zugarini, Stefano & Marco (2019)* proposed a syllable-based neural language model to generate poetry. The performance of the proposed system is relatively lower as compared to human generated poetry. In *Mukhtar & Joglekar (2021)*, ANN is used to generate the Hind and Urdu poetry. It is noted that this model also underperformed. In *Yi et al. (2018)*, *Pascual (2021)* and *Hejazi et al. (2021)* mutual reinforcement learning powered by deep leaning is used for automatic generation of poetry. The proposed models have good performance but are only applicable to specific language and can't be directly used for Pashto poetry generation. In another study, deep learning classification is used for Persain poetry (*Ruma et al., 2022*). This study focuses on specific poems of Persian language and can't be used for general poetry. In *Chy et al. (2020)*, Bengali language poetry is automatically generated using deep learning. The performance of the model is on the lower side and the model is not trained and evaluated on Pashto poetry. In *Loller-Andersen & Gambäck (2018)* poetry is generated using visual input; however, the limitation of this work is it only generates poetry using visual input. It is not able to generate poetry from text prompting.

This literature review reveals that there is lack of a suitable model for Pashto poetry generation. Thus, this study presents deep transformer based pre-trained models for automatic generation of Pashto poetry.

## METHODOLOGY

### Development of dataset

We rolled up our sleeves to explore Pashto poetry and put together a bunch of beautiful poems ourselves. We gathered these poems from more than 20 different books, written by famous Pashto poets like Abaseen Yousafzai, Shaukat Ali Shaukat, and Hussain Ahmad

Sadiq. In our collection, there is a grand total of 1,800 heartfelt ghazals. These ghazals are like emotional paintings, reflecting the diversity and depth of Pashto poetry.

Imagine each poem as a little story, told in Pashto. Some are about love, while others talk about important things happening in society and history. In these lines, there's a mix of old traditions and new ideas. Through our collection, we aim to understand how Pashto poetry has changed and what parts of its history it holds onto.

Our collection isn't just a pile of poems—it is a way to celebrate the talented poets who've given Pashto its poetic colors. Each ghazal is like a piece of art, reminding us of the language, culture, and feelings that make Pashto poetry special. By sharing these poems, we're inviting everyone to join us on a journey into the world of Pashto poetic wonders.

As shown in Fig. 1, the Pashto poetry example highlights the profound cultural significance of Pashto literature and the richness of its poetic tradition. This particular verse exemplifies the theme of love and devotion, commonly found in Pashto poetry. Additionally, it underscores the timeless nature of these poetic expressions, as they continue to resonate with Pashto-speaking audiences across generations.

As shown in Fig. 2, the following methodology to generate praise poems consists of four phases. In this section, these phases are discussed in more detail.

Figure 2 visually represents these phases, demonstrating the flow of the research methodology and its integral components. This structured approach ensures the validity and effectiveness of the research process, ultimately leading to valuable outcomes.

## Data pre-processing

In this phase, data preprocessing is performed by removing or modifying data that is incorrect, incomplete, irrelevant, or duplicate. We also removed unrelated characters or symbols, such as punctuation marks, line breaks, $, #, brackets, parentheses, and other unrelated characters from the dataset. Table 2 shows an example of the input before and after processing. During the data cleaning process for Pashto poetry, we used regular expressions (regex) in Python to identify and remove specific elements that did not conform to the desired format of the poetry. Here are a few examples of regular expressions commonly used in cleaning Pashto poetry data.

Removing non-Pashto characters: A regular expression can be used to match and eliminate any non-Pashto characters, such as punctuation marks, numbers, or special symbols. For example, the expression '[^\u0600-\u06FF\s]' matches any character outside the range of Pashto Unicode characters and whitespace. By replacing the matched characters with an empty string, the data can be cleaned by removing the unwanted elements.

Handling line breaks and extra whitespace: Pashto poetry often contains line breaks and extra whitespace that can affect the readability and format. Regular expressions can be used to address these issues. For example, the expression '\n+' matches one or more consecutive line breaks, which can be replaced with a single line break or a whitespace character. Similarly, the expression '\s+' matches one or more consecutive whitespace characters, which can be replaced with a single space.

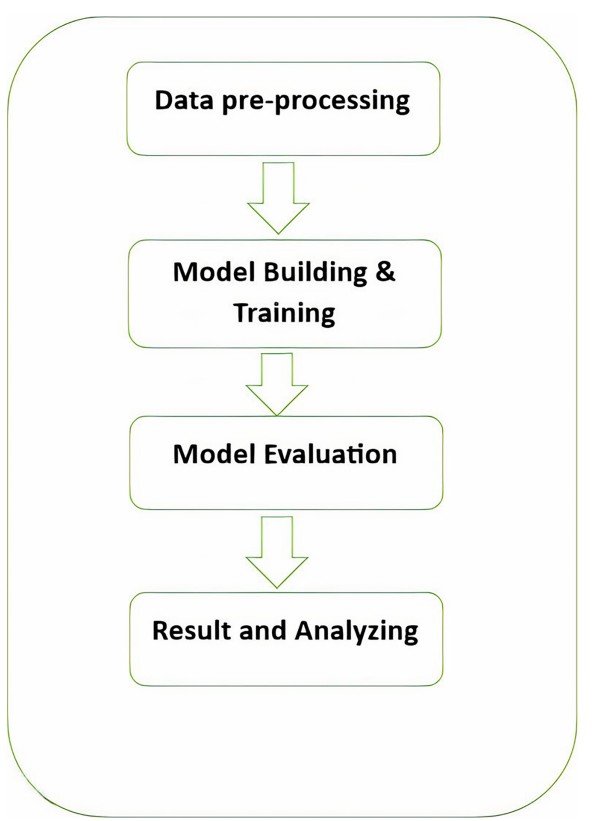

**Figure 1** **The phases of the methodology.**

دے مخكښى نه زما صنم دى وروستو نه زما غم

دے مخكښى نه زما قدم لكي نه ښكته مى پښى

داريمه يامته ډبري كه جنت زما جنت ستا

دے مخكښى نه زما غنم چي خدايه وكرمه خه

يم مخكښى نه موسم هر خوبښى د دنيا د كه زه

دے مخكښى نه زما موسم هر خوبښى د ستا كله

**Figure 2** **Pashto poetry example.**

**Table 2  Example of the pre-processing step.**

| Before processing | After processing |
|---|---|
| خندا (پوري !اکار په     ستا نه ناپوهی د راځي ___هم | راځي هم خندا پوري کار په ستا نه ناپوهی د |
| راځي" هم ژړا او //خالقه کیږي هم --می ګیله | راځي هم ژړا او خالقه کیږي هم می ګیله |
| یم ډک نه کمزورو شمبره *بي د زادیم ادم___ زه"" | یم ډک نه کمزورو شمبره بي د زادیم ادم زه |

By applying appropriate regular expressions, we were able to successfully clean the data and convert it into a specific format suitable for Pashto poetry. These expressions helped remove non-Pashto elements, address line breaks and whitespace issues, and ensure that the data adhered to the desired format of poetry. The cleaning process ensured that the resulting dataset was more coherent, standardized, and ready for subsequent analysis, such as training machine learning models for Pashto poetry generation.

Table 2 illustrates an example of the crucial data pre-processing step in the research methodology. In this phase, the raw text data undergoes several transformations to make it suitable for analysis. The 'Before processing' column displays the original, unprocessed text, which may contain various noise elements such as special characters, punctuation, and extra spaces.

In contrast, the 'After processing' column presents the text after undergoing pre-processing. During this step, unnecessary characters, symbols, and irregularities are removed or corrected, resulting in clean and structured data that can be effectively used for subsequent analysis tasks. Data pre-processing is a fundamental step in ensuring the quality and accuracy of research outcomes.

## Proposed methods

In this work, we explore two approaches for Pashto poetry generation using pre-trained models. Both approaches are instruction-based, meaning that the models are given instructions on how to generate poetry. These two pre-trained transformer models are, LaMini-Cerebras-590M and Bloomz-560m. Each model has unique capabilities and characteristics that contribute to the creative generation of Pashto poetry. After an extensive evaluation, we selected two models to proceed with further fine-tuning and finalization for our project.

Table 3 provides a detailed comparison of two proposed models, BLOOMZ-560M and LaMini-Cerebras-590M, with respect to various key characteristics.

## Big science/bloomz-560m

Big science/bloomz-560m is a large language model developed by Big Science, a distributed research project to train and deploy large language models. The model was trained on a massive dataset of text and code and can follow human instructions in dozens of languages zero-shot. To train the model, Bigscience used a process called multitask finetuning (xP3).

**Table 3 Compression table of proposed models.**

| Feature | BLOOMZ-560M | LaMini-Cerebras-590M |
|---|---|---|
| Model size | 560M parameters | 590M parameters |
| Pretraining data | 174 languages | Text and code |
| Finetuning data | xP3 in the same languages seen during pretraining | Instructions |
| Capabilities | Can follow human instructions in dozens of languages zero-shot | Can follow instructions, answer questions, generate text, translate languages, and write different kinds of creative content |
| Strengths | Good at following human instructions in multiple languages, even if it has never seen those languages before. | Versatile and can be used for a wide range of tasks. |
| Weaknesses | Not as versatile as LaMini-Cerebras-590M and can only follow instructions in languages it has been pretrained on. | Not as good at following human instructions as BLOOMZ-560M, especially in languages it has not been pretrained on. |

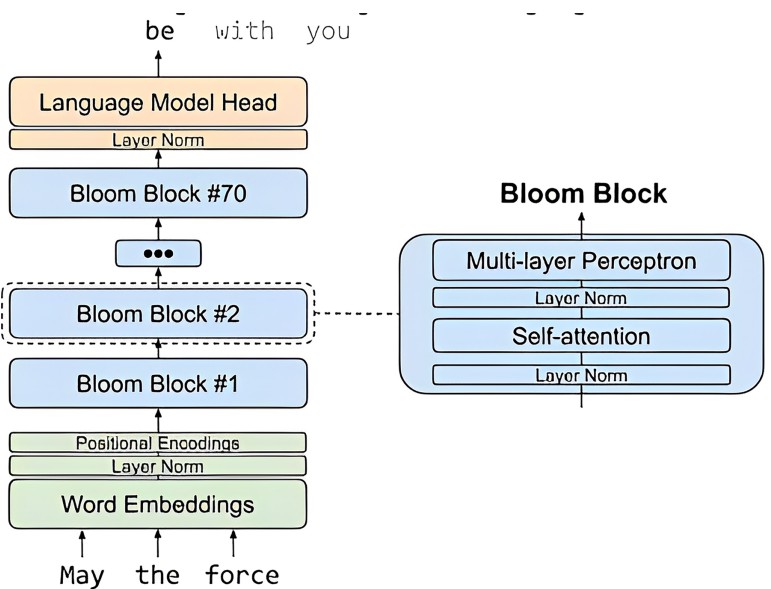

**Figure 3 Big science/bloomz-560m architecture.**

This involves training the model in a variety of tasks, such as translation, text generation summarization, and question answering. This helps the model to learn to generalize to new tasks and languages. The resulting model has 560 million parameters and is one of the largest and most powerful language models in the world.

Figure 3 depicts a high-level overview of BLOOM, a large language model (LLM) released in 2023 by Hugging Face. BLOOM is a 176B parameter model, making it one of the largest and most powerful LLMs available to the public.

Brief explanation of the key components of Fig. 3.

1) Tokenizer: The tokenizer is responsible for breaking down text into individual tokens, which can be words, subwords, or other units.

2) Encoder: The encoder takes the tokens from the tokenizer and converts them into a sequence of hidden states. Each hidden state represents the model's understanding of the token and its context.

3) Decoder: The decoder takes the hidden states from the encoder and generates text one token at a time.

4) Attention: Attention is a mechanism that allows the model to focus on specific parts of the input sequence when generating output.

The arrows in the image represent the flow of data through the model. Text is first tokenized and then passed to the encoder. The encoder generates a sequence of hidden states, which are then passed to the decoder. The decoder generates text one token at a time, using attention to focus on specific parts of the input sequence.

The Big science/bloomz-560m model was trained for Pashto poetry using Google Collab. The first step was to format the dataset into JSON format. The model was then trained using a set of parameters, including BATCH_SIZE = 128: This is the number of examples that are processed at a time. warmup_steps = 100: This is the number of steps that are used to gradually increase the learning rate. max_steps=300: This is the maximum number of steps that the model is trained for. learning_rate = 3e−4: This is the learning rate, which is a measure of how quickly the model is updated. fp16 = True: This flag tells the model to use 16-bit floats instead of 32-bit floats. This can help to improve performance. logging_steps = 10: This parameter specifies how often the model's progress is logged. After the model was trained, it was fine-tuned using a different set of parameters.

The fine-tuning parameters included: num_train_epochs = 10: This is the number of epochs that the model is fine-tuned for. weight_decay = 0.01: This is a regularization parameter that helps to prevent the model from overfitting. The fine-tuning process helped the model to learn the specific features of Pashto poetry.

As a result, the model was able to generate more creative and accurate Pashto poetry. Here is a brief explanation of the terminology used in the training process: JSON: A format for storing data in text files. Transformer: A neural network architecture that has been shown to be very effective for natural language processing tasks. Epoch: A complete pass through the training data. Batch size: The number of examples that are processed at a time. Learning rate: A measure of how quickly the model is updated. Regularization: A technique used to prevent the model from overfitting. Overfitting: A problem that occurs when the model learns the training data too well and is unable to generalize to new data. Fine-tuning: A process of adjusting the model's parameters to improve its performance on a specific task. After training the model it takes almost 40 min to train and fine-tuned for our data, they output of the train model is TrainOutput (global_step = 300, training_loss = 1.5561199736595155, metri = {'train_runtime': 2365.3552, 'train_smple_per_second': 8.117, 'train_steps_per_second': 0.127, 'total_flos': 8910943530516480.0, 'train_loss': 1.5561199736595155, 'epoch': 10.85}) parameters provided for generating poetry: max_length = 300: This parameter sets the maximum length of the generated poetry, in terms of the number of tokens (words or characters, depending on the model). do_sample = True: This parameter indicates whether to use sampling when generating poetry.

## Pashto-bloomz training Loss vs. Steps

**Figure 4** Big science/bloomz-560m training loss. 

Sampling refers to the process of randomly selecting the next token from a probability distribution, rather than always selecting the most likely token. top_k = 200: This parameter limits the number of possible next tokens to consider during sampling to the top k most likely tokens, where k is the value specified. top_p = 0.91: This parameter limits the number of possible next tokens to consider during sampling based on their cumulative probability, such that the sum of probabilities of all tokens considered is less than or equal top_p, where p is the value specified. temperature = 0.7: This parameter controls the "creativity" of the generated poetry by adjusting the randomness of the sampling process. A higher temperature value will result in more random and potentially creative output, while a lower temperature value will result in more conservative and predictable output.

Figure 4 displays the training loss of the Big Science/BLOOMZ-560M model during the training process. The y-axis represents the loss, which begins at four and gradually decreases as the model learns and converges. This loss reduction is indicative of the model's improvement in minimizing prediction errors over time.

The x-axis corresponds to the max_steps parameter, a key training argument in the Transformers library, set to a value of 300. The max_steps parameter determines the maximum number of training steps or iterations the model undergoes during the training process. It is a crucial parameter in fine-tuning models like Big Science/BLOOMZ-560M, influencing the training duration and performance.

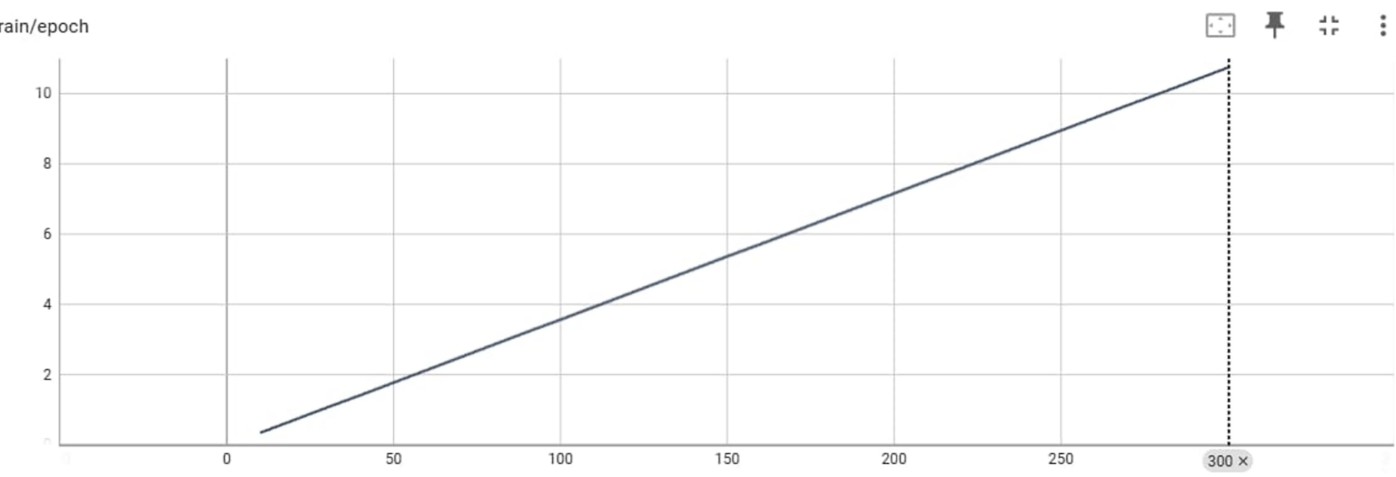

**Figure 5   Training/epoch of Bigscience/bloomz-560m.**               

Analyzing the training loss curve in Fig. 4 provides insights into the model's convergence and learning dynamics. A decreasing loss indicates that the model is progressively improving its ability to make accurate predictions and understand the underlying patterns in the training data.

Figure 5 provides an overview of the training process for the Big Science/BLOOMZ-560M model in terms of epochs and the max_steps parameter. The y-axis represents the epoch count, indicating how many complete passes the model has made through the training data. The x-axis represents the max_steps parameter, which determines the maximum number of training steps per epoch.

Training deep learning models, such as Big Science/BLOOMZ-560M, is often organized into epochs, where each epoch involves iterating over the entire training dataset once. The max_steps parameter influences the number of training steps within each epoch and, consequently, the training duration. Table 4 An Example of the Input-Output of Big Science/BLOOMZ-560M (Generated Pashto Poetry by Big Science/BLOOMZ-560M Model).

Table 5 provides a summary of the key hyperparameters used during the fine-tuning process of the Big Science/BLOOMZ-560M model.

Table 6 provides an overview of the key specifications and architecture details of the Imran1/bloom_pg model.

## MBZUAI/LaMini-Cerebras-590M

LaMini-Cerebras-590M is a large language model (LLM) developed by the Mohamed bin Zayed University of Artificial Intelligence (MBZUAI). It is a finetuned version of the Cerebras-GPT-590M model, which was trained on. The MBZUAI/LaMini-Cerebras-590M model is a large language model that was trained on a massive dataset of Pashto text. The model was trained using the Cerebras Wafer-Scale Engine, which is a supercomputer that

**Table 4 An example of the input-output of Bigscience/bloomz-560m.**

| An input sentence | An output sentence |
|---|---|
| نیول دردونه کښ وینه دی حسن تکور تکور | نیول زرونه دی ارمان هر لالیه قربان تانه |
| | دی نه می اندام یو په دی کړی خوښ می تول پریخی |
| | نیول سیندونه تول به مونږ رانیول غمی تاسو |
| | یاور هجورول نه بارود چه دی پوهان داسی |
| | نیول بمونه ي کښ اورو چه دی پوهان داسی |
| | کي قاتل ته ي خوامخواه چه دی نه دا می مطلب |
| | نیول لاسونه ي ته کور ستاسو چه کیدو مر خو |

**Table 5 Hyperparameter used in Bigscience/bloomz-560m.**

| Hyperparameters | Fine-tuning |
|---|---|
| BATCH_SIZE | 128 |
| Learning rate | 3e−4 |
| Max steps | 300 |
| Epochs | 10 |

**Table 6 Blimp architecture.**

| | |
|---|---|
| Model name | Imran1/bloom_pg |
| Model architecture | BloomForCausalLM |
| Model type | Bloom |
| Layers | 24 |
| Attention heads | 16 |
| Hidden size | 1,024 |
| Seq. length | 2,048 |
| Transformers version | 4.30.0.dev0 |
| Use cache | True |
| Vocabulary size | 250,880 |

is specifically designed for training large language models. The MBZUAI/LaMini-Cerebras-590M model can be used to generate Pashto poetry.

The Fig. 6 architecture, of MBZUAI/LaMini-Cerebras-590M model, which is a fine-tuned version of the Cerebras-GPT-590M model on the LaMini-instruction dataset. The

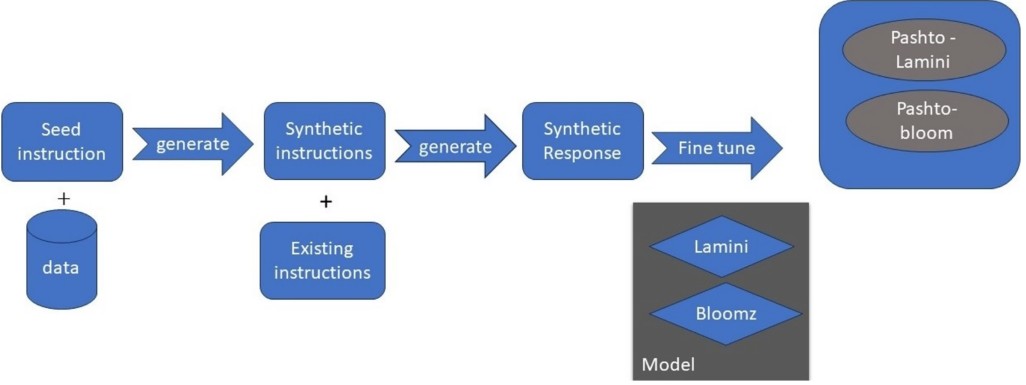

**Figure 6** **MBZUAI/LaMini-Cerebras-590M architecture.**

LaMini-instruction dataset contains 2.58M samples for instruction fine-tuning, which allows the model to learn to perform a variety of tasks, such as generating different creative text formats of text content, translating languages, writing different kinds of creative content, and answering questions in an informative way.

The MBZUAI/LaMini-Cerebras-590M architecture is a decoder-only model, which means that it does not have an encoder. This is because the model is designed to be used for instruction following tasks, where the input is a natural language instruction, and the output is the desired response.

The architecture of the MBZUAI/LaMini-Cerebras-590M model is as follows:

1) Input: The input to the model is a natural language instruction.
2) Tokenizer: The tokenizer breaks down the instruction into individual tokens, which can be words, subwords, or other units.
3) Transformer decoder: The Transformer decoder takes the tokens from the tokenizer and generates a sequence of hidden states. Each hidden state represents the model's understanding of the token and its context.
4) Output: The output of the model is the generated response to the instruction.

The Transformer decoder is a stack of self-attention layers. Self-attention is a mechanism that allows the model to focus on specific parts of the input sequence when generating output. This is important because it allows the model to understand the context of the instruction and generate a response that is relevant and informative.

The model can be used to generate a variety of Pashto poetry, including ghazals, quatrains, and sonnets. The model can also be used to generate Pashto poetry on a variety of topics, such as love, loss, and nature. The MBZUAI/LaMini-Cerebras-590M model is a valuable tool for generating Pashto poetry. The model can be used to generate creative and accurate Pashto poetry. The model is also available on the Hugging Face Hub, which makes it easy to use.

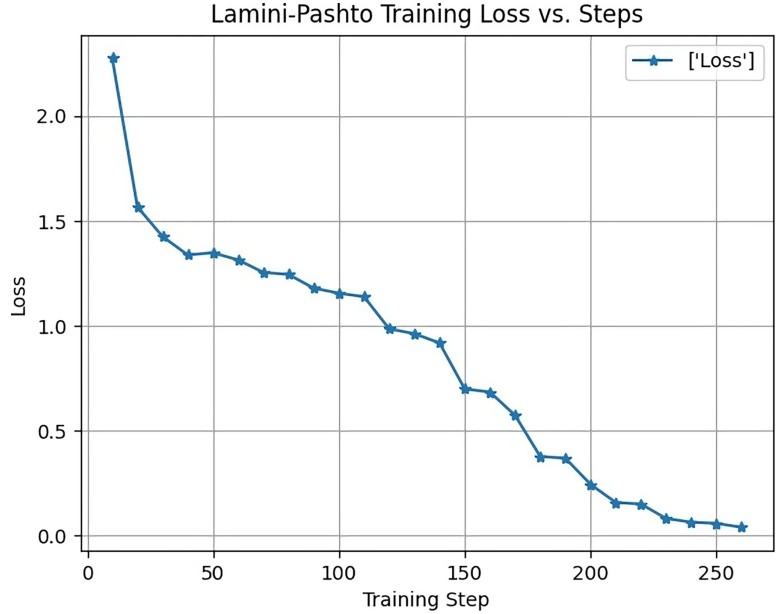

**Figure 7  Training loss of LaMini-Cerebras model.**

The MBZUAI/LaMini-Cerebras-590M model was fine-tuned for Pashto poetry generation using a JSON format dataset of Pashto poetry. The model was fine-tunegradient_accumulation_steps = GRADIENT_ACCUMULATION_STEPS: This is the number of times that the gradients are accumulated before they are used to update the model's parameters. This can help to improve performance. warmup_steps = 100: This is the number of steps that are used to gradually increase the learning rate. This can help to prevent the model from overfitting. max_steps = 260: This is the maximum number of steps that the model is fine-tuned for. learning_rate = 2e−4: This is the learning rate, which is a measure of how quickly the model is updated. fp16 = True: This flag tells the model to use 16-bit floats instead of 32-bit floats. This can help to improve performance. logging_steps = 10: This parameter specifies how often the model's progress is logged. The fine-tuning process took approximately 40 min to complete. The model was able to generate Pashto poetry that was both creative and accurate. After training the output of the train model shows that the model was trained for 260 steps, and that the loss on the training data was 0.8195405574945304. The training process took 1,708.3726 s, and the model processed 9.74 training samples per second. The model performed 0.152 training steps per second, and it performed a total of 1.3027304839053312e+16 floating-point operations during training. The model was trained for 9.4 epochs.

Figure 7 displays the training loss of the LaMini-Cerebras model during the training process. The y-axis represents the loss, which begins at 2.3 and gradually decreases as the model learns and converges. This loss reduction is indicative of the model's improvement in minimizing prediction errors over time.
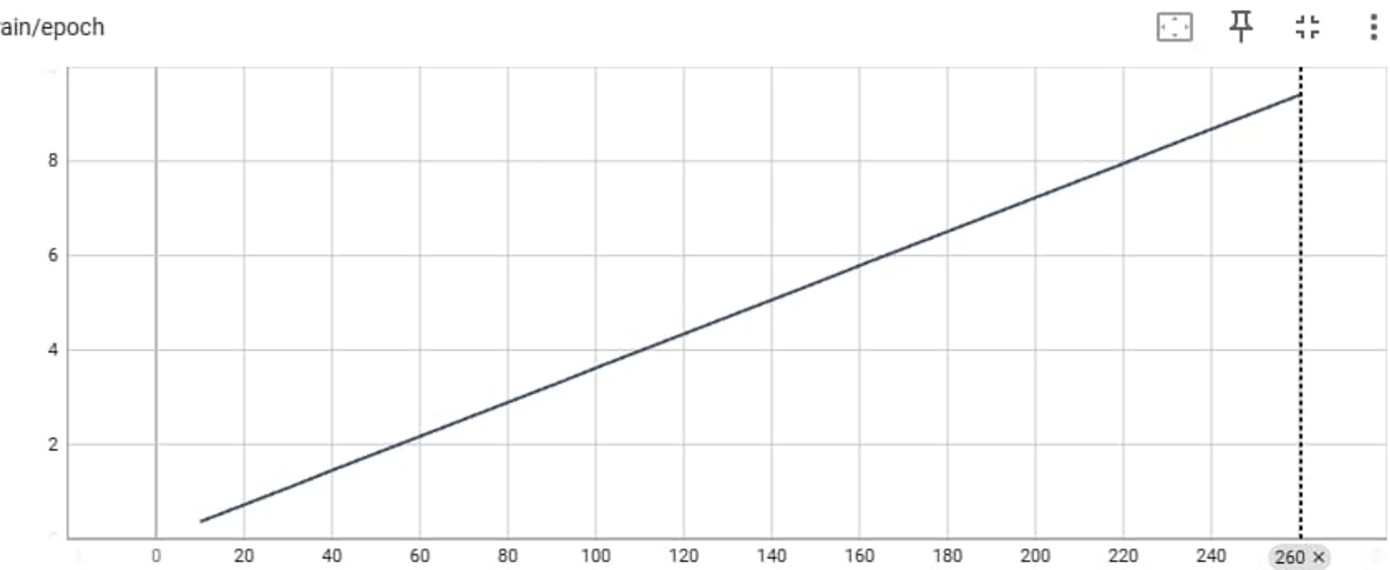

**Figure 8 Training/ephah of LaMini-Cerebras model.**

**Table 7 An example of the input-output of LaMini-Cerebras model.**

| An input sentence | An output sentence |
| --- | --- |
| نه په سر اونه په زريم | زۀ د مينى جوارگـريــم |
| | بس هم دا مى سرمايه ده چى بدنــام پـه لـروبــريــم |
| | رحدونه چى ئى نيشته زه په داسى لارو سريم |
| | چى الوت ئى تر آسمان دے هغى فكر وزريم ك |

**Table 8 Hyperparameter used in LaMini-Cerebras model.**

| Hyperparameters | Fine-tuning |
| --- | --- |
| BATCH_SIZE | 128 |
| Learning rate | 2e−4 |
| Max steps | 260 |
| Epochs | 9 |

The x-axis corresponds to the max_steps parameter, a key training argument in the Transformers library, set to a value of 250. The max_steps parameter determines the maximum number of training steps or iterations the model undergoes during the training

**Table 9 Cerebas_pg architecture.**

| | |
|---|---|
| Model name | Imran1/cerebus_pg |
| Model architecture | GPT2LMHeadModel |
| Activation function | GELU |
| Model type | GPT-2 |
| Number of layers | 18 |
| Number of attention heads | 12 |
| Number of embedding dimensions | 1,536 |
| Inner dimensions of the model | 6,144 |
| Number of positions | 2,048 |
| Transformer version | 4.28.0.dev0 |
| Use cache | False |
| Vocabulary size | 50,258 |

process. It is a crucial parameter in fine-tuning models like LaMini-Cerebras, influencing the training duration and performance.

Figure 8 provides an overview of the training process for the LaMini-Cerebras model in terms of epochs and the max_steps parameter. The y-axis represents the epoch count, indicating how many complete passes the model has made through the training data. The x-axis represents the max_steps parameter, which determines the maximum number of training steps per epoch.

Training deep learning models, such as LaMini-Cerebras is often organized into epochs, where each epoch involves iterating over the entire training dataset once. The max_steps parameter influences the number of training steps within each epoch and, consequently, the training duration. An example of the input output and the generated Pashto poetry by LaMini-Cerebras Model are shown in Table 7 below.

Table 8 provides a summary of the key hyperparameters used during the fine-tuning process of the LaMini-Cerebras 560M model.

Table 9 provides an overview of the key specifications and architecture details of the Imran1/cerebus_pg model.

# RESULTS AND DISCUSSIONS

In this section, we present the results and evaluation of the three models that were used to generate Pashto poetry.

## Quantitative evaluation

In this study, we used BLEU scores to evaluate the quality of the generated Pashto poetry, using the default parameter of 4-grams. BLEU scores are a commonly used metric to measure the similarity between the generated text and the reference text, with a higher score indicating a greater level of similarity.

**Table 10 Models output.**

| Model name | The generated poetry in Pashto | The generated poetry in English |
|---|---|---|
| Imran1/bloom_pg | تکور تکور حسن دی وینه کښ دردونه نیول<br>تانه قربان لالیه هر ارمان دی زړونه نیول | Round and round beauty captivates, but blood draws pain<br>O beloved, I sacrifice myself for you, every wish captures hearts |
| | تَول می خوښ کړی دی په یو اندام می نه دی پریخی<br>تلسو غمی رانیول مونږ به تول سیندونه نیول | I have cherished everything, not leaving a single part behind<br>You bought a gem, we would seize all the rivers |
| | داسی پوهان دی چه بارود نه یاوړ هجوړول<br>داسی پوهان دی چه اورو کښ ي بمونه نیول | Such are the wise ones who crafted explosives<br>Such are the wise ones who captured bombs in flames |
| | مطلب می دا نه دی چه خوامخواه ي ته قاتل کي<br>خو مر کیدو چه ستاسو کور ته ي لاسونه نیول | My intention is not that you become a killer in vain<br>But dying, your hands would grasp your home |
| Imran1/cerebus_pg | نه په سر اونه په زریم<br>زۀ د مینی جوارګـریـم | Neither through rank nor through gold<br>I am a merchant of love |
| | بس هم دا می سرمایه ده<br>چی بدنـام پـه لـروبـریـم | This is my only capital<br>That I am infamous in all lands |
| | رحدونه چی ئي نیښته<br>زه په داسی لارو سریم | With no boundaries<br>I wander on such paths |
| | چی الوت ئي تر آسمان دے<br>هغي فکر وزریم ک | That soar up to the sky<br>I fly on the wings of such thoughts |

**Table 11 Models BLEU score.**

| GRAM | Imran1/bloom_pg | Imran1/cerebus_pg |
|---|---|---|
| 4-GRAM (Default) | 0.57 | 0.59 |

$$\text{BLEU} = \text{BP} \cdot \left( \sum_{n=1}^{N} w_n \log p_n \right)$$

In Tables 8 and 9, we present a sample of the outputs generated by both the bloom_pg and cerebras_pg models. Our experimental results showed that both models performed well in generating new Pashto poetry that adhered to the style and tone of the poetry in our training dataset. The generated poetry was presented in a poetic format, with proper structure and rhythm, and demonstrated the models' ability to capture the underlying patterns and structures of Pashto poetry. These results are promising and demonstrate the potential for using machine and deep learning techniques to generate high-quality Pashto poetry.

Both the Imran1/bloom_pg and Imran1/cerebus_pg models have demonstrated their ability to generate high-quality Pashto poetry. As Table 10 illustrates, the generated Pashto

| Table 12 Human evaluation. | | | | |
|---|---|---|---|---|
| Criteria<br>Model name | Meaning | Coherence | Rhyme | Rhythm |
| Imran1/bloom_pg | 0.57 | 0.56 | 0.60 | 0.55 |
| Imran1/cerebus_pg | 0.51 | 0.53 | 0.57 | 0.49 |

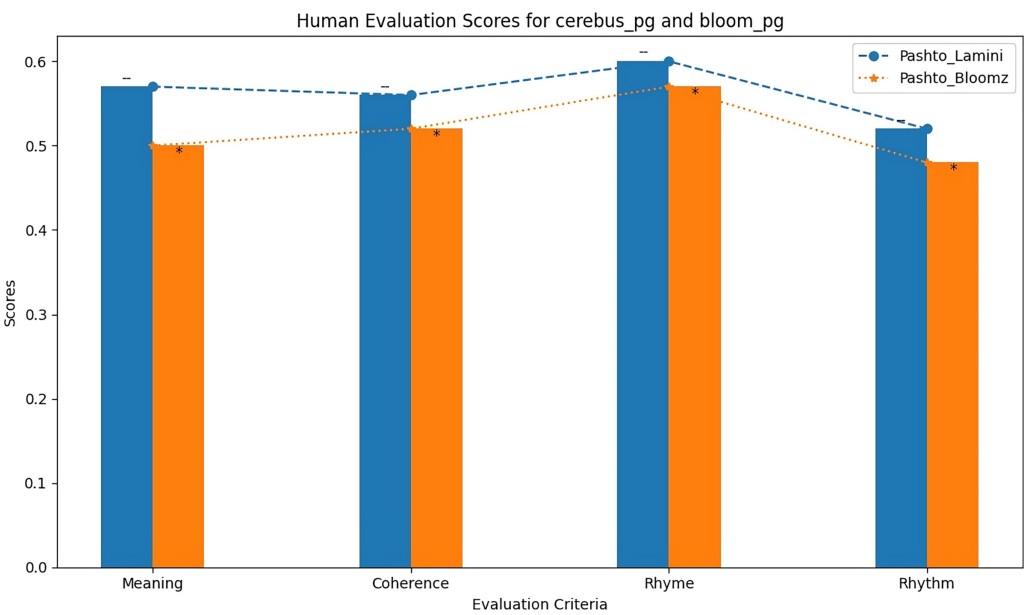

**Figure 9** **Human evaluation bar graph of both model.**

poems exhibit a rich and expressive use of the language, capturing the essence of various themes and emotions. Furthermore, we have provided manual English translations for these poems, ensuring accurate and contextually meaningful renditions, allowing for a cross-linguistic appreciation of their poetic beauty and meaning. Table 11 presents the mean BLEU-Default scores for both models. Our observation indicates that both models performed well and were able to generate novel words that preserve the tone and style of Pashto poetry written by poets.

## Qualitative evaluation

Table 12 shows the results of the human evaluation of the Imran1/cerebus_pg and Imran1/bloom_pg models based on various criteria, including meaning, coherence, rhyme, and rhythm. The scores reflect the assessment of these models in generating Pashto poetry that aligns with these criteria.

The BLEU scores alone are insufficient to evaluate the quality of generated poems, as they do not consider important factors such as meaning and coherence. Therefore, human evaluation is necessary in this field of research. However, human evaluation can be challenging due to different opinions and preferences in poetry. In this study, we randomly

selected two poems from each model and asked two experts to evaluate them based on four criteria: meaning, coherence, rhyme, and rhythm. Each criterion was scored from zero to one, with zero being the worst and one the highest score. The results of the human evaluation are presented in Table 12. As shown in the table, the Bigscience/Bloomz-560m model received higher scores in terms of meaning, coherence, and rhyme compared to the other model that is LaMini-Cerebras-590M. These findings indicate that the Bigscience/Bloomz-560m model is better at generating Pashto poetry that is not only similar to the training dataset but also the generated poetry is meaningful, coherent and have interconnectivity. The comparison of the human evaluation of the two models is also shown as bar chart in Fig. 9.

## CONCLUSIONS AND FUTURE WORK

This article presents two models for generating Pashto poetry: Bigscience/Bloomz-560m and LaMini-Cerebras-590M. Our experimental results showed that the Bigscience/Bloomz-560m model outperformed the other model based on the BLEU-4 score. However, the generated poems still lacked some grammatical rules and logical sequences of words, as well as their interconnections with each other. In the future, we plan to reduce the runtime and increase the poetry ghazals in the dataset. Additionally, we aim to improve the grammatical rules of the lines that the models generate. We believe that this research not only contributes to the preservation and promotion of Pashto literary heritage but also opens new avenues for artistic exploration and creativity.

In closing, this research provides a valuable tool for Pashto poetry enthusiasts, allowing them to generate new poetry and explore the nuances of the language with the power of machine learning and deep learning. We hope that this project inspires further research and applications in the field of computational poetry generation and contributes to the continued evolution and appreciation of Pashto literature. There are threats to validity of the models as the models are trained on massive amounts of text data. This data can contain biases and prejudices that reflect the real world.

### Funding
This research is funded by the Researchers Supporting Project Number (RSPD2024R947), King Saud University, Riyadh, Saudi Arabia. The funders had no role in study design, data collection and analysis, decision to publish, or preparation of the manuscript.

### Grant Disclosures
The following grant information was disclosed by the authors:
King Saud University, Riyadh, Saudi Arabia: RSPD2024R947.

### Competing Interests
Khursheed Aurangzeb is an Academic Editor for PeerJ.

## Author Contributions

- Imran Ullah conceived and designed the experiments, performed the experiments, analyzed the data, performed the computation work, prepared figures and/or tables, authored or reviewed drafts of the article, and approved the final draft.
- Khalil Ullah conceived and designed the experiments, performed the experiments, analyzed the data, performed the computation work, prepared figures and/or tables, authored or reviewed drafts of the article, and approved the final draft.
- Hamad Khan conceived and designed the experiments, performed the experiments, analyzed the data, performed the computation work, prepared figures and/or tables, authored or reviewed drafts of the article, and approved the final draft.
- Khursheed Aurangzeb conceived and designed the experiments, performed the experiments, analyzed the data, performed the computation work, prepared figures and/or tables, authored or reviewed drafts of the article, and approved the final draft.
- Muhammad Shahid Anwar conceived and designed the experiments, performed the experiments, analyzed the data, performed the computation work, prepared figures and/or tables, authored or reviewed drafts of the article, and approved the final draft.
- Ikram Syed conceived and designed the experiments, performed the experiments, analyzed the data, performed the computation work, prepared figures and/or tables, authored or reviewed drafts of the article, and approved the final draft.

## Data Availability

The dataset and the code are available in the Supplemental Files.

## Supplemental Information

Supplemental information for this article can be found online at http://dx.doi.org/10.7717/peerj-cs.2163#supplemental-information.

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
