# Peer review of "Pashto poetry generation: deep learning with pre-trained transformers for low-resource languages"

_PeerJ Computer Science, doi:10.7717/peerj-cs.2163_

## Round 0.1 · original submission · Major Revisions

· Academic Editor

Major Revisions

As per comments from three reviewers, I suggest a major revision for this article.

**Language Note:** The review process has identified that the English language must be improved. PeerJ can provide language editing services - please contact us at [email protected] for pricing (be sure to provide your manuscript number and title). Alternatively, you should make your own arrangements to improve the language quality and provide details in your response letter. – PeerJ Staff

·

Basic reporting

The English is better throughout the paper. Still, there are some grammatical mistakes, and it is recommended to review the paper with a fluent speaker or paid English grammar correction tools. Also, the figures need to be redrawn with high quality.

Experimental design

The research questions and methodology of the paper are well-defined. However, there are some changes recommended to further improve the quality of the paper. The recommendations are mentioned in the additional section.

Validity of the findings

The replication package is provided, and the paper's conclusion is well-stated.

Additional comments

Review:
This manuscript reviews the development of fast and computationally efficient deep transformer models for the automatic generation of Pashto poetry. The proposed models, namely LaMini-Cerebras-590M and Bloomz-560 M, have been shown to achieve high performance.
As a reviewer, I have identified several grievances and suggestions that should be addressed before considering the publication of this manuscript.
1. The abstract needs revision to show the contribution of this paper concisely and explicitly.
2. Why are only these two transformer models (LaMini-Cerebras-590M and bloomz-560m) used in this paper? More literature reviews on these methods will make it more trustworthy.
3. The organization of the paper requires attention and should be restructured. The flow of ideas and the logical sequence of sections need improvement to enhance the readability and comprehension of the manuscript.
4. The paper is recommended for proofreading and a more readable presentation. All abbreviations and variables must be declared. Also, higher quality of the figures is required.
5. Inclusion of relevant and recent papers: The references section of the manuscript requires additional relevant and recent papers to support the arguments and claims made in the paper. Expanding the reference list with more recent studies would strengthen the academic contribution of the manuscript.
6. Discussion section: In the discussion section, There is no in-depth discussion of the results to ensure new findings and guide the future study.
These suggestions and grievances, once incorporated into the manuscript, will significantly enhance its quality, and prepare it for publication.

Reviewer 2 ·

Basic reporting

1. If some data preprocessing is used then mention it in detail.
2. I noticed inconsistencies in the capitalization of the stress levels (MEDIUM, LOW, HIGH) throughout the paper. It would be helpful if the author could explain the reasoning behind the variation and ensure consistent capitalization for clarity and uniformity.
3. The organization of the paper needs improvement in terms of describing the detailed sections. Providing a clear and comprehensive overview of the content covered in each section would enhance the overall structure and facilitate better navigation for readers.

Experimental design

1. The Use of the pre-trained Transform models as fast and computationally efficient deep transfer learning (DTL) models for Pashto poetry generation require further clarification. It would be beneficial if the author could elaborate on the specific characteristics or features that contribute to this description. Providing more details about the speed and computational efficiency of these models would enhance the readers' understanding.

Validity of the findings

1. The paper uses BLEU score for performance evaluation. Therefore, the authors must provide some explanation for selecting these metrics and their relevance to automatic poetry generations. If there is another evaluation metrics possible then also use that so the reader of the manuscript is more clear about the model performance.
2. The authors need to highlight existing research gaps and limitations in the submitted work and address the generalization of the proposed models to the poetry of different poets.
In conclusion, while the paper presents a valuable topic, the authors are advised to address the concerns to enhance the overall quality and impact of the paper.

Reviewer 3 ·

Basic reporting

This manuscript introduces two models, Bigscience/Bloomz-560m and LaMini-Cerebras-590M, for generating Pashto poetry. The study reveals that the Bigscience/Bloomz-560m model demonstrates superior performance over its counterpart, as evidenced by its higher BLEU-4 score. However, the paper's readability is challenging, and it is recommended that the authors undertake thorough proofreading to enhance clarity and coherence.

Experimental design

The methodology is straightforward, with a sufficient detailing of hyper-parameters, suggesting that replication of the study could be feasibly achieved.

Validity of the findings

Impact and novelty not assessed. The experiments does not support well on the hypothesis.

Additional comments

A notable limitation of the paper is its reliance solely on fine-tuning existing datasets, which detracts from its originality. The study primarily adapts pre-existing fine-tuning techniques to a poetry dataset. To augment the paper's contribution, the authors should consider expanding their future work. This could include efforts to reduce runtime, augment the number of ghazals in the dataset, and refine the grammatical structure of the poetry generated by the models. Such enhancements would not only bolster the paper's originality but also its practical applicability and academic relevance.

---

## Round 0.2 · Minor Revisions

· Academic Editor

Minor Revisions

As per comments from two original reviewers, there are still a few minor issues need to be fixed, so I suggest a minor revision.

·

Basic reporting

However, the authors have improved the English of the paper. Still, there are a few minor grammatical mistakes that need to be improved. Authors can take the help of native speakers or can use existing grammar correction tools.

Experimental design

The experimental design of the paper has been enhanced and manuscript looks better now.

Validity of the findings

The manuscript is refined and now looks good and concise for the readers to gain the main idea. However, the following section may be looked at again for typos/misspellings if any. The introduction section and Literature review are good enough, but if possible, add the worth of automatic Pashto poetry application in the introduction section.

1- Please add the applications of the models in the Introduction Section
2- Have a look at general for typos if any in the manuscript.
3- The contribution of the paper should be listed in the bulleted form. It would help readers find out the key contributions of the paper.
4- Add a concise statement about the novelty of the work.
5- The related work is quite large, It would improve the readability of the paper if the author could group the related work into several sub-sections
6- Also, authors haven't discussed possible threats to validity in the paper

Reviewer 3 ·

Basic reporting

Literature references, sufficient field background/context provided.

Experimental design

Research question well defined, relevant & meaningful. It is stated how research fills an identified knowledge gap.

Validity of the findings

All underlying data have been provided; they are robust, statistically sound, & controlled.

Additional comments

This revision has been improved a lot. I think it is good now.

---

## Round 0.3 · accepted · Accept

· Academic Editor

Accept

In the opinions of reviewers and mine, this revised paper can be accepted.

·

Basic reporting

The Paper presentation has been improved comprehensively.

Experimental design

The experimental results are improved now. All the details are incorporated in the paper about experimental design.

Validity of the findings

The approach addresses low-resource language issues,i.e., pashtoo, which is a timely and interesting area to work on, as not many literature works exist previously.